# Assessment of Piezoelectric Sensors for the Acquisition of Steady Melt Pressures in Polymer Extrusion

**Sónia Costa, Paulo F. Teixeira \*, José A. Covas \* and Loic Hilliou \***

Institute for Polymers and Composites/i3N, University of Minho, 4800-058 Guimarães, Portugal; sonia_25_21-09@hotmail.com

\* Correspondence: p.teixeira@dep.uminho.pt (P.F.T.); jcovas@dep.uminho.pt (J.A.C.); loic@dep.uminho.pt (L.H.); Tel.: +351-253510320 (L.H.)

**Abstract:** Piezoelectric sensors have made their way into polymer processing and rheometry applications, in particular when small pressure changes with very fast dynamics are to be measured. However, no validation of their use for steady shear rheometry is available in the literature. Here, a rheological slit die was designed and constructed to allow for the direct comparison of pressure data measured with conventional and piezoelectric transducers. The calibration of piezoelectric sensors is presented together with a methodology to correct the data from the inherent signal drift, which is shown to be temperature and pressure independent. Flow curves are measured for polymers showing different levels of viscoelasticity. Piezoelectric slit rheometry is validated and its advantage for the rheology of thermodegradable materials with viscosity below 100 Pa·s is highlighted.

**Keywords:** piezoelectric; pressure transducers; extrusion; rheology

## 1. Introduction

Accurate readings of steady melt pressure are important to monitor flow conditions during polymer extrusion and for rheometry experiments involving pressure flows. For example, it is well-known that the installation of flush-mounted pressure gauges along a slit die wall allows the assessment of shear viscosity up to relatively high shear stresses, with no need of the Bagley correction for non-viscometric flow effects at the edges of the die [1]. However, the conventional melt pressure transducers of the diaphragm type are bulky and have large front dimensions (typically a diameter of 7.8 mm and process connections with M18 ($\times$ 1.5) according to DIN 3852-1592, or 1/2-20 UNF), which may impact on the practical possibility of fixing them on small and/or curved flow channels. In the case of slits, consecutive transducers along the length maybe set apart more than desired and there may be little space for additional transducers. Indeed, if a microscope is also inserted in the slit, it is possible to image the structure of the material undergoing shear flow, thereby giving way to rheo-optical characterization [2]. Piezoelectric sensors have remarkable sensitivity, fast response and reduced size compared to conventional melt pressure sensors. They are often used in injection molding to measure instantaneous pressures along the production cycle. When mounted in customized miniaturized systems (such as capillary rheometry dies), and implementing oversampling techniques, piezoelectric sensors were used to establish relationships between rapid pressure fluctuations with small amplitudes and distortions found on the surface of the extruded materials [3–9]. As a result of these studies, slit dies equipped with piezoelectric transducers are now commercially available as accessories to capillary rheometers. Using these devices, it was possible to predict extrusion instabilities for a series of commercial polyethylenes [10]. A similar piezoelectric set-up was coupled to a laboratory screw

extruder to demonstrate the feasibility of applying high sensitivity detection systems to practical polymer processing [6].

Unlike conventional diaphragm sensors, piezoelectric sensors exhibit a drift of the signal with time. The drift is inherent to both the charge amplification and transducer-to-amplifier cabling, which are needed to convert the transducer signal into volts. Therefore, it is necessary to adopt data treatment procedures to account for the drift of the piezoelectric signal and for the non-linearity of the transducer with pressure [11] or temperature [12]. Recently Kádár et al. [5] used piezoelectric sensors in a slit die attached to a capillary rheometer to estimate the first normal stress difference of polymer melts, $N_1$, via the so-called 'pressure hole effect' [13]. The method is very demanding for the pressure transducers, as small differences between the readings of two sensors are to be measured [14]. Kádár et al. [5] assumed the drift in the voltage $V_{drift}$ to be a linear function of time $t$,

$$V_{drift}(t) = V_0 + st, \tag{1}$$

where $V_0$ is the value of the voltage at time 0 defining the start of the recording, and $s$ is the slope that depends on the testing temperature and the pressure applied on the piezoelectric transducer. Eventually, a flow curve was obtained which compared reasonably well with small amplitude oscillatory shear data in a Cox-Merz representation, but a single slope was used to correct the signal drift recorded during the application of a ramp of steady shear rate steps, which correspondingly generated a ramp of pressures.

A direct comparison between the flow curves measured with piezoelectric sensors and conventional pressure transducers during steady shear has not yet been reported in the literature. Thus, the use of piezoelectric sensors for steady shear rheometry remains to be validated. This work proposes a methodology to correct the drift together with temperature and pressure effects. A modular slit die was designed and constructed in order to incorporate both piezoelectric sensors and conventional pressure transducers in a mirror-like arrangement along the channel, thus enabling the direct comparison between the flow curves acquired with the two types of transducers.

## 2. Experimental

### 2.1. Materials

An extrusion grade low density Polyethylene (ALCUDIA® LDPE 2221FG, from Repsol, Spain) with a density of 0.922 g/cm³, a melt flow index of 2.1 g/10 min (190 °C/ 2.16 kg) and a processing temperature range between 150–180 °C was used for the assessment. This polymer has excellent thermal stability, thus minimizing the eventual influence of thermal degradation in the experiments. Two biodegradable polymers, a polyhydroxybutyrate (PHB P309 from Biomer®, Krailling, Germany) and a Polybutylene adipate terephthalate (PBAT, ecoflex® F Blend C1200, from BASF, Ludwigshafen, Germany) were also used to compare the operability windows of conventional and piezoelectric transducers.

### 2.2. Experimental Set-Up

A double slit rheometrical die for in-process characterization, recently developed by the authors [15], was modified and used in this study. It includes three modules: a central body (module 1) where the inlet circular channel from the extruder is progressively converted into a slit, which is then divided into perpendicular measurement (module 2) and extrusion (module 3) channels. The flow rate in the measuring slit is varied by a valve positioned at its entrance, whereas a second valve balances the flow rate in the extrusion slit to keep a constant pressure at the die inlet. Thus, rheometrical measurements can be performed while maintaining constant extrusion conditions (feed rate and screw speed). The initial design, construction, and rheological validation have been reported elsewhere [15]. For the present study, a new measurement channel (module 2) was developed. The new module comprises two halves bolted together. In one part, three conventional melt pressure transducers flush

mounted can be inserted, while in the other part three piezoelectric transducers can be also flush mounted at the same axial position, see Figure 1. The measurement channel presents a cross section of 10 mm width by 0.8 mm height.

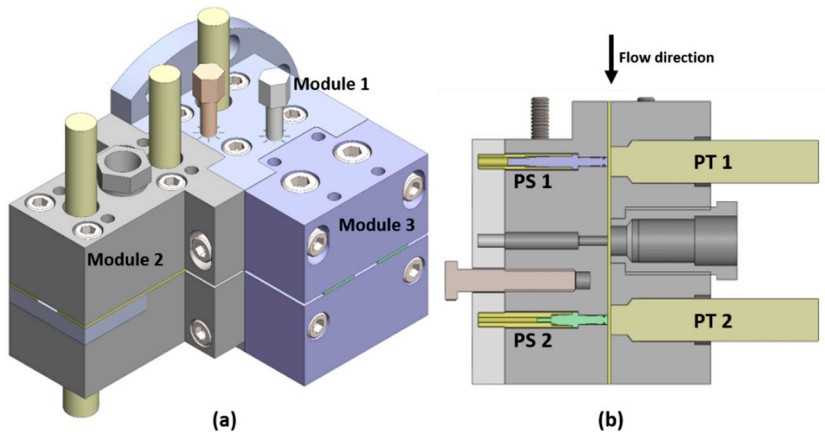

**Figure 1.** Schematic view of the double slit rheometrical die (**a**) and cross section view of the measurement channel (module 2) (**b**).

Two piezoelectric transducers from the Kistler Group, Winterthur, Switzerland (Kistler 6182B (PS 1) and a Kistler 6189A (PS 2), see main characteristics as provided by the manufacturers in Table 1) were used. PS 2 has the ability to perform simultaneously pressure and temperature measurements and was mounted closer to the die exit. In order to minimize the signal drift, as well as the influence of external electromagnetic noise, the sensors were separately shielded by means of a copper mesh covering the wires. Two conventional pressure transducers Dynisco PT422A (0-3000 PSI, Dynisco Inc., Franklin, MA, USA) with a sensitivity of ±0.5% and a front diameter of 7.8 mm were used (PT 1 mounted upstream and PT 2 mounted downstream). These transducers are sensitive to variations in the temperature in the range of ±0.005 MPa for ±1 °C. All sensors were flush mounted in the double slit die and adequate fixing was verified.

**Table 1.** Characteristics of the melt pressure transducers.

| Characteristics | Kistler 6182 B | Kistler 6189A | PT422A |
|---|---|---|---|
| Front diameter | 2.5 (mm) | 2.5 (mm) | 7.8 (mm) |
| Range | 0–200 (MPa) | 0–200 (MPa) | 0–21 (MPa) |
| Sensibility, $x$ | −2.5 (pC/bar) | −6.6 (pC/bar) | 0.5% |
| Operating temperature range: | | | |
| Sensor, cable, connector box | 0–00 (°C) | 0–200 (°C) | |
| At the front of the sensor | < 450 (°C) | < 450 (°C) | 0–400 (°C) |

As a validation strategy, and in order to avoid the inherent fluctuations of pressure and flow rate in screw extruders, the double slit die was attached on the barrel of a Rosand RH10 capillary rheometer (Malvern Instruments, Malvern, UK), see Figure 2.

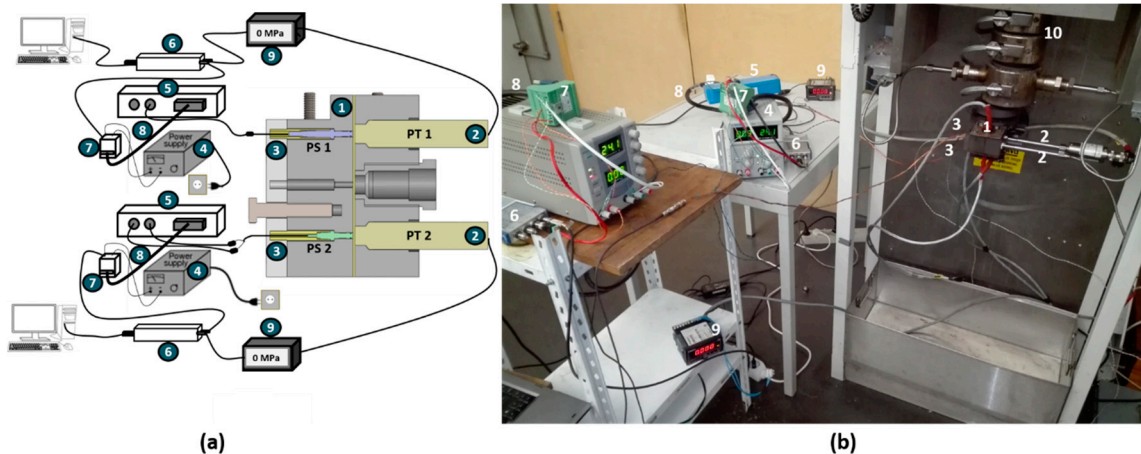

**Figure 2.** Schematic view (**a**) and photograph (**b**) of the experimental set-up. 1: double slit die. 2: conventional pressure transducers. 3: piezoelectric sensors. 4: power supply. 5: charge amplifier. 6: analog-to-digital converter (ADC) card. 7: terminal block for 25-Pin. 8:D-SUB Cable. 9: strain gage indicator. 10: capillary rheometer.

The double slit die is independently heated using two temperature controllers OMRON E5CSV (Omron Corporation, Tokyo, Japan). In order to acquire the signals from the piezoelectric sensors, the following equipment was used:

- Two charge amplifiers, a Kistler 5155A 2241 for PS 1 and a Kistler 5155A 22A1 (with separate channels to acquire pressure and temperature) for PS 2. Two amplification ranges are available which depend on the maximum charge delivered by the transducer, namely up to 20000 pC (which corresponds to 10 V in Range I) and up to 5000 pC (which corresponds to 10 V in Range II for larger amplification).
- Two power supplies (Matrix MPS-5LK-2, delivering a voltage between 18–30 V DC) to feed the amplifiers and command functions.
- Two analog-to-digital converter (ADC) boards (NI-9215 from National Instruments) driven by custom-written LabVIEW™ routines to digitize the amplifiers outputs and enhance the sensor sensitivity with on-the-fly oversampling techniques [16–18].
- Two DIN-Rail Mount Terminal Blocks for 25-Pin D-SUB Modules and two 25-Pin Shielded D-SUB cables (from National Instruments, Austin, TX, USA) in order to interface the power supply, the amplifier, and the ADC.

Each conventional pressure transducer was connected to a Dynisco 1390 strain gage indicator, with analog retransmission output accuracy span of ±0.2%. The strain gage indicator is also connected to a data acquisition system NI-9215 from National Instruments (see Figure 2) and driven by custom-written LabVIEW™ routines. The piezoelectric sensor PS 1 and the conventional pressure transducer PT 1 were connected to the same data acquisition system NI-9215 while PS 2 and PT 2 were connected to the second data acquisition system NI-9215, for time synchronization.

*2.3. Experimental Procedure*

2.3.1. Calibrations

Conventional transducers and piezoelectric transducers were coupled to a Terwin T1200 Mkll hydraulic comparison test pump. Calibration curves were constructed by reporting the pressure returned by the transducers as a function of the pressure set by the test pump. For the piezoelectric

transducers, the calibration was performed with the charge amplifiers set in the Range II. In this case, the voltage returned by the transducers are converted into pressure using the following equation:

$$PS\ 1 = PS\ 2 = (V_{cor} * 50)/x \tag{2}$$

where $x$ is the sensibility given by the manufacturer (see Table 1) and $V_{cor}$ is the output voltage $V(t)$ corrected from the drift, namely $V_{cor}(t) = V(t) - V_{drift}(t)$.

### 2.3.2. Pressure Measurements and Flow Curves

The whole set-up is switched on at least 30 min prior to any measurement to allow for a stabilized temperature in both capillary barrel and double slit die, and to warm up the electronics, therefore reducing the signal drift. The slit is then fed with melt (maintaining constant the piston velocity) for a period of 1 min and the material is left to relax until the pressure readings, PT 1 and PT 2, stabilize. Then, the zero balance of the conventional transducers is adjusted in order to define a zero pressure and to adjust the 80% span. The data acquisition by the ADC starts and the amplifier is switched on to record the piezoelectric drift $V_{drift}(t)$ over two minutes. This time is needed for measuring a consistent slope $s$. Afterwards, 6 successive incremental step increases of the piston velocity are performed in order to ramp up the corresponding shear rates. In each ramp the piston velocity varied from 10 mm/min to 60 mm/min, corresponding to apparent shear rates ranging from 27 s$^{-1}$ to 165 s$^{-1}$. The shear rate was step increased only after steady state pressures were read from the graphic display provided under the LabVIEW™ environment. At the end of the ramp, the piston is retracted and the drift monitored for two minutes. The flow curves were constructed using the following analysis for slit rheometry [1], where the pressure drop $dP$ = PS 1 − PS2 or $dP$ = PT 1 − PT 2 and the volumetric flow rate, $Q$, are used to determine the wall shear stress $\sigma$ and the shear rate $\dot{\gamma}$, respectively, with the following equations:

$$\sigma = \frac{H}{2(1 + H/W)} \frac{dP}{dx} \tag{3}$$

where $W$ and $H$ are the width and height of the flow channel respectively, and $dx$ is the distance between transducers PS 1 and PS 2 or PT 1 and PT 2. The apparent shear rate $\dot{\gamma}_a$ is:

$$\dot{\gamma}_a = \frac{6Q}{WH^2} \tag{4}$$

where $Q$ is obtained in an indirect way by measuring the weight of the extrudate. This methodology is preferred to the use of the piston speed for $Q$ determination, as it can readily be extended to in-line rheometry during extrusion application. The true wall shear rate was calculated from:

$$\dot{\gamma} = \frac{\dot{\gamma}_a}{3}\left(2 + \frac{d\ln\dot{\gamma}_a}{d\ln\sigma}\right) \tag{5}$$

## 3. Results and Discussion

Figure 3 presents the time evolution of the voltage $V$ acquired by transducer PS 1 mounted in an empty slit (no pressure applied), in order to assess the drift inherent to this measuring system. In Figure 3a, the effect of switching on the amplifier on the recorded voltage is evident: the signal jumps to a value $V_0$ before drifting with time. As expected [5], the signal presents a drift that follows Equation (1). Thus, the voltage can be corrected to give a flat signal $V_{cor}(t) = V(t) - V_{drift}(t)$, as shown in the inset to Figure 3a. Both jump to $V_0$ and drift are due to the closing of the electronic circuit (both amplification and pressure sensing), and as such should be independent of the pressure and temperature at the tip of the sensor [19].

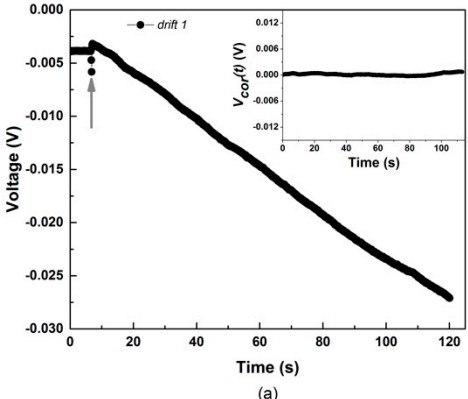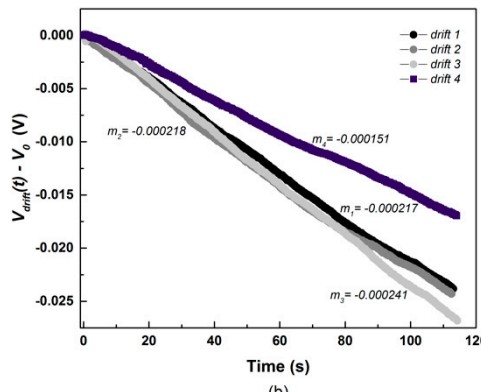

(a)　　　　　　　　　　　　　　　　　　　　　(b)

**Figure 3.** Drifts measurements with piezoelectric sensor PS 1 in an empty slit. (**a**) Time dependence of the voltage delivered by the charge amplifier. The arrow indicates the time at which the amplifier is switched on. The inset in (**a**) displays the corrected voltage $V_{cor}(t)$. (**b**) Reproducibility of the slope of the drift $V_{drift}(t) - V_0$.

Figure 3b shows the measurements of three consecutive independent drift measurements (drift 1, 2, and 3) and a drift measurement performed after 3 h (drift 4). The slope *s* does not change significantly in the three consecutive measurements, in fact there is almost an overlap of the data. Drift 4 deviates slightly from the previous measurements, but it reflects only a difference of 0.0019 MPa from the average of the consecutive drifts (using the sensibility of the manufacturer in the conversion). In order to check for any effect of pressure on $V_{drift}(t)$, the piezoelectric sensors were individually mounted on a calibrator and several pressures were successively applied. Figure 4 reports the time evolution of the output voltage $V(t)$ during the ramping up of pressure in the calibration pump. The inset in Figure 4 displays the pressure dependence of the slope *s*, recorded with sensor PS 1. The error bar associated to each data point results from the error computed from the fit of Equation (1) to $V(t)$ in each pressure step. The data show that in the applied pressure range, there is no significant effect of the pressure on the drift of the piezoelectric sensor.

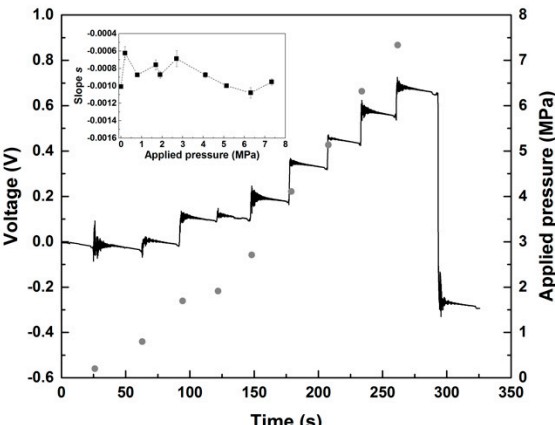

**Figure 4.** Time evolution of the output voltage $V(t)$ (solid line) of the PS 1 sensor during the ramping up of pressure (circles) in the calibration pump. The inset displays the pressure dependence of the slope *s* for the PS 1.

Figure 5 represents the calibration curves for PS 1 and PS 2. Each data point and error bar result respectively from the average and standard error of five measurements similar to the one displayed in Figure 4. The slopes returned by the linear fits to the data indicate that the response of the piezoelectric sensors is nicely linear for the range of pressures tested. In addition, the computed slopes do not significantly differ from the constants calculated with the sensibilities reported in the calibration

certificate supplied by the manufacturer, namely $50/x = 20$ for PS 1 and $50/x = 7.58$ for PS 2 (see Table 1 for $x$). The intercepts to the origin for the linear fits are $0.28 \pm 0.03$ and $0.096 \pm 0.001$ for PS1 and PS2, respectively. These intercepts are smaller than those found for the conventional transducers ($0.415 \pm 0.015$), which suggest a better sensitivity for piezoelectric sensors in the low-pressure range.

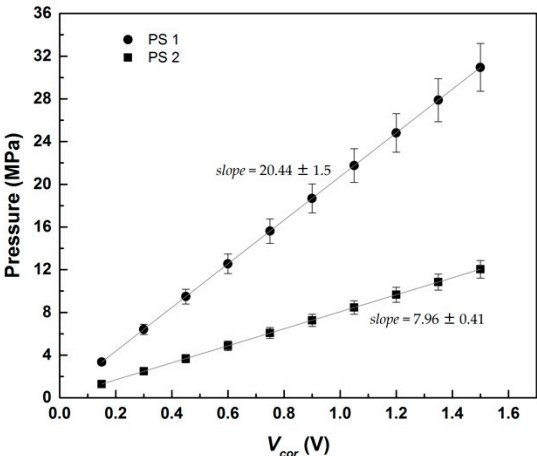

**Figure 5.** Calibration curves for PS 1 and PS 2, fitted with a linear function.

In order to estimate the eventual effect of the melt temperature on $V_{drift}$, experiments were performed with the piezoelectric transducers mounted on the heated double slit die, but with no material in the slit cavity. Figure 6 reports the temperature dependence of the slope $s$ computed using Equation (1) from $V_{drift}$ transients recorded with PS 2.

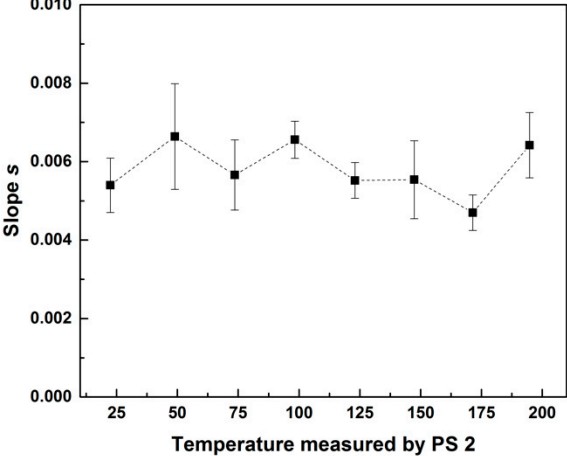

**Figure 6.** Average drift slope $s$ as function of temperature.

Each point and error bar in the figure correspond to an average value and error computed from five measurements. For the temperature range tested in Figure 6, which corresponds to the range recommended by the manufacturer, no noteworthy variation of $s$ with temperature is perceived. This result, together with the data displayed in Figure 4, confirms that $V_{drift}$ is essentially due to the electronic imperfections of the whole piezoelectric acquisition system [11,12,19].

Figure 7 presents the time evolution of the outputs of the piezoelectric and conventional transducers recorded during the ramping up of piston velocities.

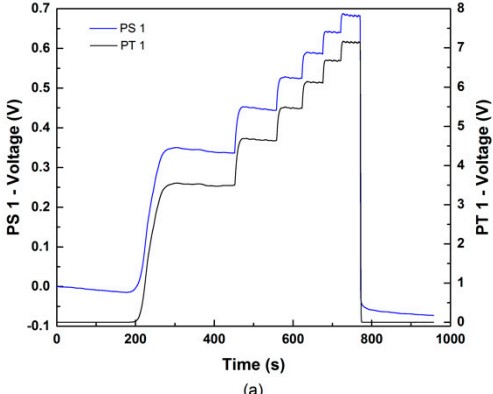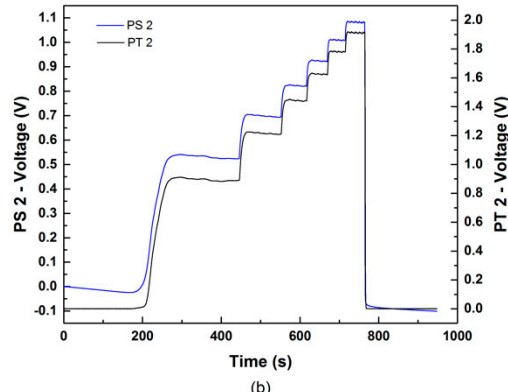

(a) (b)

**Figure 7.** Time evolution of outputs (voltages) measured during the stepwise ramp in the piston velocity of the capillary rheometer fed with a low density polyethylene (LDPE) at 150 °C. (**a**)—Piezoelectric (PS 1) and conventional (PT 1) transducers located upstream. (**b**)—Piezoelectric (PS 2) and conventional (PT 2) transducers located downstream.

The double slit rheometer was fed with LDPE at 150 °C. The transients of face-mounted transducers show a satisfactory matching along the time scale. The piezoelectric signals show the expected drifts before the inception of melt flow, and after stopping and retracting the piston of the capillary rheometer. The piezoelectric data were corrected by computing $V_{cor}(t)$ using either $V_{drift}(t)$ measured before actuating the piston or $V_{drift}(t)$ measured at the end of the ramp. The steady state values of $V_{cor}(t)$ recorded for each piston velocity were then converted into pressure using the calibration curves, see Figure 5. The resulting pressures PS 1 and PS 2 are plotted in Figure 8 as a function of the piston velocities, together with the pressures PT 1 and PT 2 measured with the conventional transducers. Experiments performed at 180 °C are also reported in Figure 8. Each curve in Figure 8 is the result of the average of two ramps performed to check for data reproducibility. The corresponding error bars for each data point are smaller than the symbols used to represent the data. The error resulting from the reproducibility is larger (ranging from 0.02% to 3.4% for piezoelectric sensors and 0.02% to 1.2% for conventional transducers) than the error from the pressure reading (ranging from 0.13% to 0.3% for piezoelectric sensors and 0.16% to 0.5% for conventional transducers).

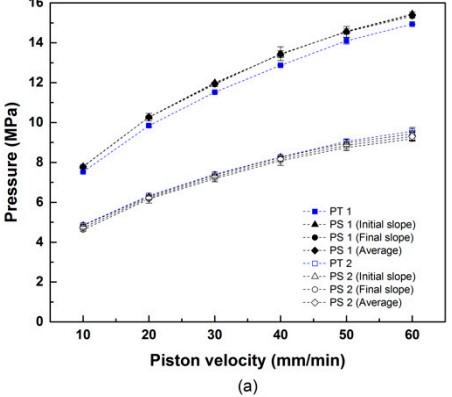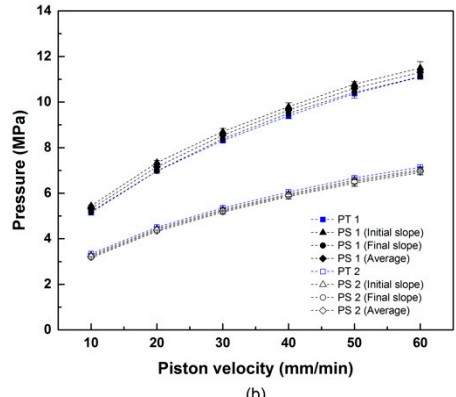

(a) (b)

**Figure 8.** Comparison between the pressures obtained with the piezoelectric sensors and conventional transducers for each piston velocity. (**a**) LDPE at 150 °C. (**b**) LDPE at 180 °C. Initial slope and final slope refer to the values of *s* used in the fitting of $V_{drift}$ before actuating the piston or at the end of the ramp in piston's velocity. Average indicates that drift correction is performed by computing the average of initial and final slopes *s*.

Data in Figure 8 indicate a moderate mismatch between the pressures returned by the two types of transducers, albeit their contour being virtually identical. Piezoelectric sensors measured somewhat

larger values than the conventional sensors upstream, whereas the opposite occurs downstream. Generally, the method of drift correction (*s* fitted at the beginning or at the end of the ramp, or *s* computed from the average of these two values) does not impact significantly on the results, as the mismatch between conventional and piezoelectric measurements remains in the range of 0.4% to 4% (see Table 2). This conclusion is consistent with the results displayed in Figures 4 and 6 where the drift is shown to be independent of both pressure and temperature. Overall, the numbers reported in Table 2 compare well with the 2% scatter reported in the pressure measurements of a conventional transducer used for the capillary rheometry of a low-density polyethylene at 150 °C [20].

**Table 2.** Differences (in %) between the pressures measured with piezoelectric (PS) and conventional transducers (PT), and between the corresponding computed viscosities.

| Temperature | Sensors | Initial Slope | | Final Slope | | Average Slope | |
|---|---|---|---|---|---|---|---|
| | | Pressure | Viscosity | Pressure | Viscosity | Pressure | Viscosity |
| 150 °C | PT 1 vs. PS 1 | 2.4–3.0% | 6.7–9.3% | 1.9–3.1% | 9.1–11% | 2.2–3.0% | 8.0–10% |
| | PT 2 vs. PS 2 | 0.4–1.0% | | 1.6–3.6% | | 0.8–2.0% | |
| 180 °C | PT 1 vs. PS 1 | 2.5–4.0% | 6.7–12% | 0.0–0.8% | 5.0–7.1% | 1.3–2.3% | 5.6–9.2% |
| | PT 2 vs. PS 2 | 1.0–1.5% | | 2.3–3.9% | | 1.7–2.7% | |

The data with the average slope method for drift correction displayed in Figure 8 were inserted in Equations (3)–(5) to compute the flow curves shown in Figure 9. The impact of the mismatch between the pressure readings of both type of transducers on the resulting flow curves is evident in Figure 9. The pressure mismatch produces a vertical shift between the flow curves, the differences between the measured viscosities being larger at smaller shear rates than at larger shear rates. Nonetheless, the discrepancy between the measured viscosities remains in the range 5.6% to 10% (see also Table 2), which can be assumed as acceptable if one considers the 10% experimental error usually reported for rotational rheometry [21] and recently confirmed with Newtonian viscosity standards [22] and with a low density polyethylene studied at 150 °C [14].

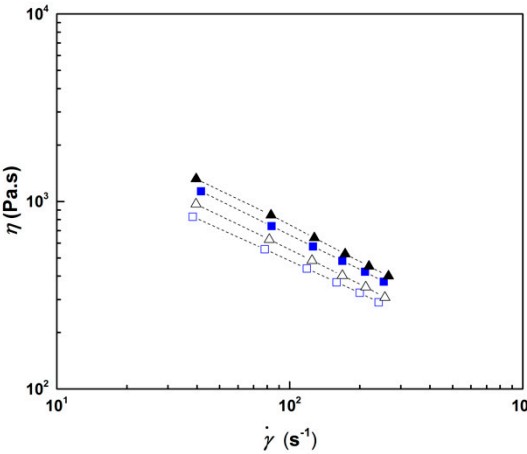

**Figure 9.** Flow curves measured with LDPE at 150 °C (solid symbols) and 180 °C (open symbols) using the piezoelectric (triangles) and the conventional (squares) pressure transducers.

The operability window of the piezoelectric transducers is now examined by testing materials showing different levels of viscoelasticity and comparing the resulting flow curves with those measured using conventional transducers. All drift corrections were performed using the average between the slopes recorded before and after the ramp in piston velocities. A PBAT grade designed for film blowing application was selected for exploring the upper viscoelastic part of the operability window, as this biodegradable material shows significant melt strength. The flow curves were measured at 190 °C and are presented in Figure 10. Overall, both sets of data, acquired either with conventional or piezoelectric

transducers with an equally high level of precision (error bars are smaller than the symbols size), nicely overlap. This result validates the use of piezoelectric transducers for slit rheometry, at least for materials showing shear viscosities in the range 500–1000 Pa·s at shear rates between 20 s$^{-1}$ and 200 s$^{-1}$. A bio-based and compostable PHB was selected for screening low viscosity materials. PHB is well known for its limited thermal stability and processability. As such, this is a challenging material for capillary rheometry, and flow curves of polyhydroxyalkanoates are scarcely found in the literature [23–25]. Indeed, true wall shear rates could not be computed from the data since steady state flow was not achieved for any piston velocity (see the inset in Figure 10). Thus, the apparent viscosity, $\eta_a$, is reported in Figure 10 as a function of the apparent shear rate, $\dot{\gamma}_a$, for the two flow curves measured at 190 °C. In spite of these difficulties, a monotonic shear thinning flow curve with satisfactory error bars is reported for the experiment carried out with the piezo transducers up to a shear rate of 150 s$^{-1}$. In contrast, larger error bars are obtained with the conventional transducers and the flow curve displays an unrealistic shear thickening at larger shear rates. Thus, this result confirms the good performance of piezoelectric transducers for detecting small pressure variations [3–6]. The slope of the PHB flow curve measured with the piezoelectric sensors is larger than −1, which suggests the occurrence of possible flow instabilities. However, the piezoelectric transient in the inset to Figure 10 does not show oscillations with frequency and amplitude signaling the presence flow instabilities as reported elsewhere [7]. Therefore, the steep slope in the PHB flow curve seems related to the fact that uncorrected apparent viscosities and shear rates are reported in the graph. In addition, the effect of thermal degradation on the shear viscosity further contributes to an apparent shear thinning.

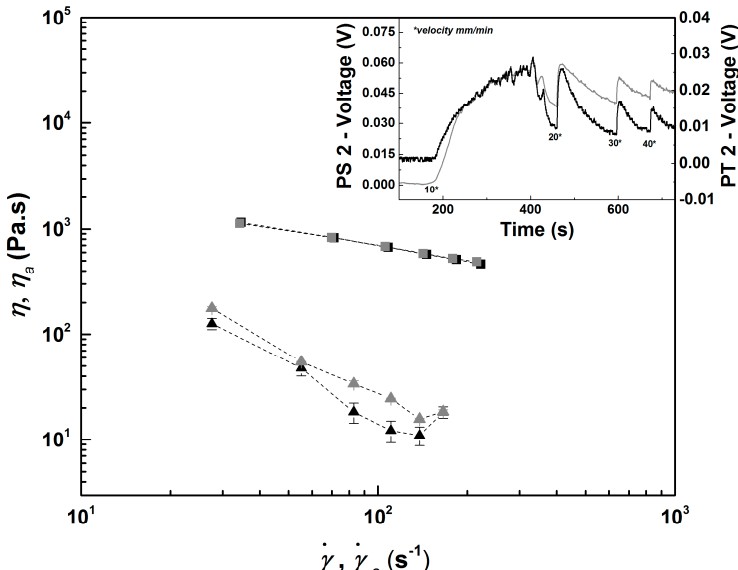

**Figure 10.** Flow curves measured with polybutylene adipate terephthalate (PBAT) at 180 °C (squares) and polyhydroxybutyrate (PHB) at 190 °C (triangles). Comparison between piezoelectric sensors (symbols and lines in grey) and conventional pressure transducers (symbols and lines in black). The inset represents the time evolution of outputs (voltages) measured with PHB by sensors located upstream during the stepwise ramp in the piston velocity of the capillary rheometer.

## 4. Conclusions

The use of piezoelectric transducers for steady melt pressure measurement has been assessed by directly comparing the flow curves measured with such transducers and with diaphragm transducers conventionally used in capillary rheometry and in polymer extrusion. A modular slit die was designed and constructed to allow for the direct comparison between the pressure data acquired by both types of transducers. Piezoelectric transducers were first calibrated and the drift inherent to the sensors electronics was analyzed as a function of both temperature and pressure. Various methodologies for

the drift correction were proposed and the results confirmed that the drift does not depend on the temperature nor on the pressure. The pressure data acquired with piezoelectric transducers differ from the data returned by conventional transducers. The difference ranges from 0% to 4% for all shear rates and temperatures tested with a LDPE. Differences in pressure readings result in a maximum 10% difference in the shear viscosities measured with LDPE for all experimental conditions tested. This difference is acceptable given the 10% error usually reported for viscosities measured either with rotational rheometers or capillary rheometers. Accordingly, the results reported here validate the use of piezoelectric transducers for slit rheometry. A better agreement between the shear viscosities measured with the two types of transducers was achieved with a more elastic PBAT. In contrast to this, piezoelectric transducers outperformed conventional transducers for measuring the steady shear viscosity of a biodegradable PHB with values of the order of 20 Pa·s at shear rates of 100 s$^{-1}$.

**Author Contributions:** The conceptualization of the whole study was performed by P.F.T., J.A.C., and L.H. Experiments were performed by P.F.T. and S.C. P.F.T., J.A.C. and L.H. analyzed the data. P.F.T. drafted the paper and P.F.T., J.A.C., and L.H. wrote, reviewed and edited the paper.

**Funding:** This research was funded by National Funds through FCT—Portuguese Foundation for Science and Technology, Reference UID/CTM/50025/2013 and FEDER funds through the COMPETE 2020 Programme under the project number POCI-01-0145-FEDER 007688. L.H. acknowledges funding from the FCT Investigator Programme through grant IF/00606/2014.

**Acknowledgments:** The authors are grateful to Miguel Gomes and Jasper Cooremans from the Polymer Engineering Department of University of Minho for support in the set-up of experiments.

**Conflicts of Interest:** The authors declare no conflict of interest.

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
