# Peer review of "Assessment of Piezoelectric Sensors for the Acquisition of Steady Melt Pressures in Polymer Extrusion"

_fluids, doi:10.3390/fluids4020066_

Round 1
Reviewer 1 Report
The manuscript “Assessment of piezoelectric sensors for the acquisition of steady melt pressures in polymer extrusion” by S. Costa et. al., is an interesting paper where the authors investigate the use of piezoelectric transducers for steady melt pressure measurement. The authors propose a modular double slit die for the assessment of the two different pressure measurement system.
The manuscript is well written and well-articulated, and the methodology used for the experimental adequately addresses all the issues encountered.
I recommend the article to be published after minor revisions:
Figure 2: this figure is not sufficiently explanatory of the apparatus used for the measurements. Please, make sure each part of the system is well visible and clearly identifiable.
Author Response
Reviewer: Figure 2: this figure is not sufficiently explanatory of the apparatus used for the measurements. Please, make sure each part of the system is well visible and clearly identifiable.
Answer:To help identifying better each part of the experimental system, Figure 2 is now augmented with a scheme (Fig 2a) which support the picture in the original Figure 2 which is now Figure 2b.
Reviewer 2 Report
The paper describes an experimental comparison between piezoelectric and standard pressure transducers for the measurements of shear viscosity in slit die rheometers.
The paper is well written, showing a good level of innovation and usefulness. The results are well presented and discussed and the conclusions are drawn with clarity.
In summary I recommend publication of the paper, provided one major and a couple of minor issues are considered:
Major point. Lines 298-299. Figure 10. In the figure, the experimental data for PHB show a too steep slope (steeper than -1). This implies that the stress is a decreasing function of the shear rate, which is always an indication of flow instabilities. Did authors notice this point? They should comment on this.
Minor points:
line 163-165: Why did the authors measure the flow rate by measuring the weight of extrudate (which requires the melt density) instead of using the piston speed?
line 204. Figure 5. What is the complite fitting equation for the transducer calibration? Are the straight lines crossing the origin? If not, how large is the offset? In other words, the authors should comment on the sensitivity of their transducers in the low pressure range
Author Response
Reviewer 2: Major point. Lines 298-299. Figure 10. In the figure, the experimental data for PHB show a too steep slope (steeper than -1). This implies that the stress is a decreasing function of the shear rate, which is always an indication of flow instabilities. Did authors notice this point? They should comment on this.
ANSWER: The following lines are now added in the revised manuscript after line 297, to comment the steep slopes for PHB data in Figure 10: “The slope of the PHB flow curve measured with the piezoelectric sensors is larger than -1, which suggests the occurrence of possible flow instabilities. However, the piezoelectric transient in the inset to Figure 10 does not show oscillations with frequency and amplitude signaling the presence flow instabilities as reported elsewhere [7]. Therefore, the steep slope in the PHB flow curve seems related to the fact that uncorrected apparent viscosities and shear rates are reported in the graph, and to the additional effect of thermal degradation on the shear viscosity”.
Reviewer 2: line 163-165: Why did the authors measure the flow rate by measuring the weight of extrudate (which requires the melt density) instead of using the piston speed?
ANSWER: Since the whole in-line set-up is intended to be implemented on extrusion lines where no piston speed is available to compute flow rates, the weighting of extrudates was the preferred method. The following line is now added to line 165 to address the raised question: “This methodology is preferred to the use of the piston speed for Q determination, as it can readily be extended to in-line rheometry during extrusion application.”
Reviewer 2: line 204. Figure 5. What is the complite fitting equation for the transducer calibration? Are the straight lines crossing the origin? If not, how large is the offset? In other words, the authors should comment on the sensitivity of their transducers in the low pressure range
ANSWER: The following lines are now added after line 204 to address the transducers sensitivity in the low pressure range: “The intercepts to the origin for the linear fits are 0.28 ± 0.03 and 0.096 ± 0.001 for PS1 and PS2, respectively. These intercepts are smaller than those found for the conventional transducers (0.415 ± 0.015), which suggest a better sensitivity for piezoelectric sensors in the low pressure range”.
Round 2
Reviewer 2 Report
The manuscript can be accepted in its present form